# Gender inequality in work location, childcare and work-life balance: Phase-specific differences throughout the COVID-19 pandemic

**Mara A. Yerkes**[1,2]*, **Janna Besamusca**[1], **Roos van der Zwan**[3], **Stéfanie André**[4‡], **Chantal Remery**[5‡], **Ilse Peeters**[1]

1 Department of Interdisciplinary Social Science, Utrecht University, Utrecht, The Netherlands, 2 The Centre for Social Development in Africa, Johannesburg, South Africa, 3 Netherlands Interdisciplinary Demographic Institute, KNAW/University of Groningen, The Hague, The Netherlands, 4 Department of Public Administration, Radboud University, Nijmegen, The Netherlands, 5 Department of Economics, Utrecht University, Utrecht, The Netherlands

☉ These authors contributed equally to this work.
‡ These authors also contributed equally to this work.
* M.A.Yerkes@uu.nl

**Data Availability Statement:** There are legal restrictions to sharing the minimal data for this study publicly. The data used for analyses are owned by the LISS panel and stored in the LISS

## Abstract

### Objective

Much research on the early stages of the COVID-19 pandemic demonstrates the unequal impact on men and women in many countries but empirical evidence on later stages of the pandemic remains limited. The objective of this paper is to study differences between men and women in work location, the relative division of childcare, and perceived work-life balance across and throughout different phases of the pandemic using six waves of probability-based survey data collected in the Netherlands between April 2020 and April 2022 (including retrospective pre-pandemic measures).

### Method

The study used descriptive methods (longitudinal crosstabulations) and multivariate modelling (cross-sectional multinomial logits, with and without moderators) in a repeated cross-sectional design.

### Results

Results suggest the pandemic is associated with several phase-specific differences between men and women in where they worked and their relative division of childcare in the Netherlands. Men were less likely than women to work fully from home at the start of each lockdown and to work on location during the first lockdown. Amongst parents, fathers increased their share of childcare throughout the first phase of the pandemic, and this increase remains visible at the end of the pandemic. Women in the Netherlands did not

data archive. Any person interested in replicating the study analyses can request access to the LISS data archive and download the data. Subsequently, all steps necessary to reproduce the analyses (i.e., the syntax files) with these data files are outlined in .txt files, all of which are available on the project website (www.cogisnl.eu/dataandsyntax).

**Funding:** Research material for the Covid19 Gender (In)equality Survey Netherlands (COGIS-NL) study was supported by an ODISSEI (Open Data Infrastructure for Social Science and Economic Innovations) grant (no funding number) to collect data during the COVID-19 pandemic. All authors listed on the paper except Ilse Peeters were recipients of the grant (MY, JB, RvdZ, SA, CR). The grant did not provide direct financing but rather allowed data to be collected within the existing LISS panel (the Dutch Longitudinal Internet Studies for the Social Sciences). The LISS panel data (including the COGIS-NL data) are collected by CentERdata (Tilburg University, The Netherlands) through its MESS project funded by the Netherlands Organization for Scientific Research. The authors do not receive funding from either CentERdata or the Netherlands Organization for Scientific Research for the purposes of this study. The funder had no role in study design, data collection and analysis, decision to publish, or preparation of the manuscript. CentERdata provided advice regarding study design for the purposes of survey programming; final decisions on study design were the responsibility of the authors. CentERdata had no role in the data analysis, decision to publish, or preparation of the manuscript. Additional funding was provided by Utrecht University (Faculty of Social and Behavioural Sciences; Faculty of Economics), the Department of Public Affairs at Radboud University Nijmegen and the Department of Sociology at the University of Amsterdam.

**Competing interests:** No authors have competing interests.

experience worse work-life balance than men throughout the pandemic, but mothers did experience worse work-life balance than fathers at various points during the pandemic.

## Discussion

Our results suggest varying long-term implications for gender inequality in society. Gender differences in work location raise concerns about the possible longer-term impact on gender inequalities in career development. Our findings on childcare suggest that many households have experienced different divisions of childcare at different stages of the pandemic, with some potential for longer-term change.

## Conclusion

Inequalities between men and women in work, childcare, and wellbeing have neither been alleviated by nor unilaterally worsened during the COVID-19 pandemic.

## Introduction

The COVID-19 pandemic and government containment measures to reduce the spread of the Sars-Cov2 virus created concerns in most industrialized countries that women and men would be impacted differently in paid work [1,2], unpaid work (e.g., childcare) [3], and wellbeing [4], thereby exacerbating existing gender inequalities (i.e., structural inequalities between men and women based on the uneven distribution of resources and burdens in combination with the structural undervaluation of women's position in society).

Although national labour markets were differentially affected by the pandemic across countries [5,6] and the consequences for workers (e.g., job loss, work hours reduction, loss of wages) varied dependent on government policies such as job retention and unemployment schemes [7,8], women were more likely than men to be affected by pandemic-induced changes to paid work during the first year of the pandemic [9,10]. For example, women were more likely than men to work on location given their overrepresentation in many, but certainly not all, essential occupations [11]. Overall, although women were generally hit harder by the economic consequences of COVID-19 [1], for some, their overrepresentation in essential occupations counterbalanced job losses experienced by workers in non-essential occupations [12].

The pandemic also differentially affected men and women in relation to household divisions of childcare. Some fathers increased their participation in childcare tasks, leading to smaller inequalities between mothers and fathers at the start of the pandemic, for example in the US, Canada, Germany and the Netherlands [13–17]. Other studies suggested the pandemic had a significantly more negative impact on women than men, for example because some women took up a larger share of childcare and home schooling tasks, were more likely to change the days or times they worked to accommodate childcare, or because despite fathers' increased involvement, mothers continued to do more childcare [e.g., 15–20].

The pandemic also led to a significant decline in wellbeing during the initial stages, particularly for mothers [e.g., 21,22]. Some studies suggest mothers' wellbeing decreased more than fathers' due to increased childcare responsibilities or changed working conditions [e.g., 21]. Work-life balance, which can be seen as a form of wellbeing, also decreased more for women than for men at the start of the pandemic [e.g., 23], although effects differ across countries. For example, in a study comparing tertiary educated women in Finland and the Netherlands, part-

time work appeared to protect wellbeing. Finnish mothers reported significantly lower work-life balance than Dutch mothers during the first lockdown, with part-time work being more widely available in the Netherlands [22]. Similar results have been found in Germany, where part-time work led to improved life satisfaction among women, but only during the first lockdown [24].

The empirical evidence from the first months of the pandemic provided essential insights into differential effects of the pandemic for men and women in paid work, childcare, and wellbeing. Yet this same evidence is challenged by the use of cross-sectional convenience samples in many countries and the limited analyses on these developments in later phases of the pandemic. The few studies available on later phases seems to suggest that the increase in inequalities is likely dependent on pandemic phases and on the country context, given variation in which containment measures were in force [14,25–27]. In the UK, for example, changes towards more gender equal divisions of housework all but disappeared among couples with children once lockdown measures were lifted [25]. Similarly, in the Netherlands, within the first year of the pandemic, as containment measures relaxed, the initial increase of fathers' relative share in childcare declined [26,27].

Our aim is to provide an empirical contribution by studying the full two years of the pandemic, thereby disentangling the short- and middle-term effects of the pandemic on gender inequality in three key areas: work, childcare, and wellbeing. These three axes of gender inequality (in paid work, unpaid work (e.g., childcare), and wellbeing) were some of the most persistent prior to the pandemic in industrialized countries [28–31] and were expected to be impacted greatly by pandemic containment measures [e.g., 32,33].

At the start of the pandemic, we established that gender inequalities among parents with co-resident minor children were evident in only certain areas of paid work, childcare and wellbeing during the first lockdown in the Netherlands [15]. Using six waves of representative, probability-based longitudinal panel data from the Netherlands (with a total of seven time points, including retrospective pre-pandemic data), we now empirically assess differences between men and women with and without co-resident minor children within two-person, different-sex households between April 2020 (the first Dutch lockdown) and April 2022. We focus on work location as an important aspect of paid work during the pandemic, the relative division of childcare between mothers and fathers, and self-perceived work-life balance (as a measure of wellbeing). We research the extent to which men and women differ in work location, the relative division of childcare, and work-life balance at different stages of the pandemic and assess if and how gender inequalities in these three domains developed throughout the pandemic in the Netherlands.

## The potential for sensemaking and the Dutch context

The now expansive pandemic literature on gender inequalities, together with extant pre-pandemic literature on gender inequalities in paid work, unpaid work (e.g., childcare), and wellbeing, suggest multiple theoretical mechanisms offer potential explanations for understanding gender inequalities in work location, the relative division of childcare, and work-life balance during the pandemic (for a detailed overview, see [77]). Some of the most commonly applied mechanisms include time availability (i.e., that men and women divide care tasks based on the time available alongside paid work), resources perspectives (i.e., that differences between men and women in paid or unpaid work reflect differences in bargaining positions based on absolute (own) or relative (the partner's) resources, such as income and education), and the 'doing gender' perspective (i.e., that women and men's behaviour reflects varying ways of reaffirming or countering societal expectations of men and women).

Although such perspectives offer useful insights for studying specific relationships (e.g., gender differences in paid work hours, or the relative distribution of housework), our focus here is broader. We therefore draw on the sociological idea of sensemaking, that is, the process by which people derive meaning from collective experiences [34]. In this manner, people can 'make sense' or give structure to the unknown [35], especially during crisis situations in which normal rules and procedures no longer apply or are no longer in place [36].

The initial lockdown and uncertainty people faced at the beginning of the pandemic could be seen as a 'cosmological episode' [34] in which workers and employers were unsure about how to respond to the situation. This uncertainty creates potential for changes in patterns of work location and childcare and, related to this, changes in work-life balance. From a sensemaking perspective, the largest uncertainties arose at the beginning of the pandemic and at the start of new lockdowns. At these stages, we would expect the largest changes in inequalities around work location, the relative division of childcare, and perceived work-life balance. Hence, we expect the greatest opportunity for a decline or risk for an increase in gender inequalities in work location, relative division of childcare and work-life balance to be during the first and subsequent lockdowns in the Netherlands. In the long run, however, as people make sense out of this pandemic situation and find new routines for themselves and their approach to work and family, we expect gender inequalities to return to pre-pandemic levels.

How men and women make sense of the uncertainty faced during the pandemic and what this means for patterns of gender inequality, can be further shaped by intersections with other social categories. Extant pre-pandemic studies highlight, for example, the relationship between utilizing flexible work arrangements, such as working from home, and parental gender roles (i.e., societal expectations of mothers and fathers; [37]), suggesting intersections between parenthood and gender. In more traditional societies, where women are expected to take on greater caregiving roles, women are more likely to use time gained from working from home to care more, whereas men are more likely to experience an increase in leisure time [38], thus suggesting potentially different drivers of gendered experiences of work-life balance among people with and without co-resident minor children. Pandemic-based literature also suggests that gendered differences in work, childcare, and wellbeing are potentially stratified by educational level (e.g., the fact that tertiary educated workers were more likely to work from home; [15]) as well as occupation, particularly in relation to whether occupations were categorized as 'essential' [e.g., 39,40]. We therefore explore the phase-specific nature of gender differences and whether gender inequalities in work, childcare, and wellbeing, experienced throughout the pandemic, are moderated by parenthood, working in essential occupations, or education differences.

As suggested above, the potential for changes between pre-pandemic gendered patterns in work, childcare, and wellbeing deriving from the need to make sense of the pandemic situation were likely greatest at the start of the pandemic and any further lockdowns. Thus, the potential for sensemaking depends on the pandemic context and government containment measures. The Netherlands entered an initial lockdown in mid-March 2020 (see S1 Fig for a timeline of relevant pandemic measures in the Netherlands). Containment measures were focused on optimizing public health outcomes, while simultaneously keeping the economy functioning and reducing the impact on society [15]. Compared to other countries, containment measures were relatively mild at this stage [41].

A containment measure with potentially broad ramifications for gender inequality was the mandate to work from home. From 12 March to early September 2020, employees were urgently requested to work from home where possible. Exceptions were made only for essential occupations, a distinction categorized by the government. Initially, essential occupations included health care (including youth care and social support), formal childcare, public

transport, the food chain (e.g. supermarkets), transport industry, waste/garbage collection and processing, media and communication, education, emergency services, and necessary government processes. Women were overrepresented in many of these occupations, particularly in the education and care sectors [42]. The Dutch government adopted quite generous policies to protect workers during the pandemic, allowing employers to maintain wage payments and avoid lay-offs, mostly in sectors where little or no work was possible (e.g., restaurants and catering). Measures were also taken to protect the self-employed. In part due to these measures, workers in the Netherlands were relatively less likely to face severe reductions of working hours than workers in other countries [6,43].

A key measure with the potential to shape gender inequality among parents was the full closure of schools and childcare centres [44]. This was a national measure, with no regional or local differences. Primary schools were completely closed through mid-May 2020, and did not completely re-open until June 8th (school ends annually in mid- to late-July). Limited school and childcare services were available to parents in essential occupations [44], yet use of such services was limited during this first lockdown. As a result, the large majority of children (88%) was home-schooled during the first lockdown [15]. When schools reopened, many children still could not fully attend schools and day care centres coping with staff shortages due to COVID-19 infections and quarantine rules [15]. In addition, the use of grandparent care, which is an important alternative to formal childcare for Dutch parents [45], was strongly discouraged to prevent infection among the older population.

Despite an initial relaxation of measures throughout the summer of 2020, following an increase in infections containment measures were once again enacted from September 2020 onwards [46], culminating in a second lockdown from mid-December 2020-early February 2021. In advance of the lockdown, people were advised to stay at home (including for work purposes) as much as possible from 4 November 2020 onwards. Schools and childcare centres were closed from 14 December onwards. Primary schools did not reopen until early February; secondary schools followed early March. Limited emergency school and childcare services were by now in high demand but difficult to obtain [47]. This second lockdown was more restrictive than the first one and included an evening curfew, in place until late April 2021. A third and final lockdown took place between December 2021-January 2022, with a work-from-home mandate already taking effect on 12 November 2021. Schools and childcare centres closed mid-December and re-opened relatively quickly in January 2022. The final containment measures were lifted in March 2022, just prior to the final data collection for our study.

## Materials and methods

### Data collection

This article draws on the COVID-19 Gender Inequality Survey Netherlands (CoGIS-NL) study, a longitudinal panel study of the differences between men and women in paid work, unpaid work, and wellbeing, carried out by the authors in the Netherlands between April 2020 –April 2022. Survey data were collected through the Longitudinal Internet Studies for the Social Sciences (LISS) panel (administered by CentERdata, Tilburg University, the Netherlands), a representative, probability-based panel derived from register data from Statistics Netherlands that is administered monthly.

### Sample

The initial scope of the CoGIS-NL study at wave 1 (fielded April 2020) included all LISS panel members in a household with at least one member in paid employment and at least one co-resident minor child (i.e., under the age of 18). These initial sampling criteria were broadened at

**Table 1. Sample statistics for each wave, including response rates.**

|  | Fieldwork | Response rate (%) | Sample (N) | Final analytical sample (N) |
|---|---|---|---|---|
| Wave 1 | April 2020 | 70.0 | 852 | 680 |
| Wave 2 | July 2020 | 75.7 | 1220 | 828 |
| Wave 3 | September 2020 | 78.8 | 1239 | 851 |
| Wave 4 | November 2020 | 74.8 | 1081 | 740 |
| Wave 5 | November 2021 | 79.0 | 1084 | 746 |
| Wave 6 | April-May 2022 | 79.0 | 1024 | 704 |

Notes: Response rates based on completed surveys. Final sample may differ slightly from the number of completed surveys reported in the codebooks because respondents who did not meet study inclusion criteria (the respondent or partner is employed and for wave 1 at least one co-resident minor child) were excluded. The increase in sample size between waves 1 and 2 reflects the addition of a sample of individuals without co-resident minor children. Data collected in July 2020 included retrospective information for the month of June.

wave 2 (June 2020; fielded July 2020) to include similar individuals without co-resident minor children. Therefore, an additional sample of respondents was included at wave 2. Sampling criteria for this additional sample were all LISS panel members in a household with at least one member in paid employment, within two standard deviations of the average age of respondents from the initial sample in wave 1 (resulting in respondents aged 28 to 57 years old). In short, wave 1 data included only respondents with co-resident minor children; data from waves 2–6 included respondents both with and without co-resident minor children. Response rates across the waves ranged from 70% to 79%. The sample at each wave ranged from 852 to 1239 respondents, dependent on the wave being studied (see Table 1 for an overview of sample statistics and fieldwork data for all waves). Study data were matched with individual-level data from two core LISS modules: "Family and Household" (collected annually in September) and "Work and Schooling" (collected annually in April). Data collection during the first wave coincided with the first lockdown in April 2020; subsequent waves were collected in July 2020 (with retrospective data from June), September 2020, November 2020, November 2021 and April 2022 (see S1 Fig for a timeline of data collection in relation to pandemic measures).

The final analytical sample for our analyses (see final column, Table 1) was reached following the exclusion of respondents who were unemployed or did not live with a partner (see S1 Table). Unemployed respondents were excluded here given the focus on work location and work-life balance. Although in some country contexts, like the US, women were disproportionately affected by pandemic-related unemployment compared to men, this was much less the case in the Netherlands [48,49]. Few unemployed respondents in our sample were unemployed due to the pandemic (between 0% and 4%; measured waves 1 through 4). Respondents without a partner were excluded given our focus on the relative division of childcare. We further excluded respondents with missing values on the covariates. Finally, respondents without co-resident minor children were excluded from the analyses on the relative division of childcare. The final analytical sample at each wave ranged from 645 to 819 respondents. See S1 Table in the supplemental file for detailed information about missing values by wave. Additionally, for an overview of descriptive statistics for all variables of interest, see S2 Table.

## Study design

In each wave, the survey questionnaire contained items measuring multiple aspects of respondents' participation in paid work (e.g., when and where they worked), the relative division of childcare and household tasks, and wellbeing (e.g., self-perceived work-life balance,

relationship satisfaction, stress). In addition, in the first questionnaire completed by a panel respondent, retrospective items on these same topics were asked to obtain baseline, pre-pandemic measures. These baseline questions were administered to respondents with co-resident minor children at wave 1, and to the respondents without co-resident minor children at wave 2. All questionnaires were administered in Dutch. In each subsequent wave, the questionnaire was adapted slightly (e.g., to adjust for changes during the pandemic). The codebooks of waves 1 and 2 of data collection, including all questions and response categories, are available from the Longitudinal Internet Studies for the Social Sciences (LISS) panel archive [50] and the study website (www.cogisnl.eu). The codebooks of waves 3 to 6 are available from the authors upon request.

## Ethical considerations

Ethical approval for data collection rests with CentERdata, the administrator of the LISS-panel. All LISS respondents are required to provide online informed consent before participating in the panel. In addition, the CoGIS-NL study was evaluated and received ethical clearance from the Ethical Assessment Board of the Faculty of Social and Behavioural Sciences, Utrecht University (approval number: 20–269). All data belong to the LISS panel archive and the authors had no access to information that could identify individual participants during or after data collection.

## Measurements

**Outcome variables.** Our analysis included three outcome variables: work location, the relative division of childcare, and work-life balance. Work location was measured with the question "What best describes your **current** work situation?" Answer categories included: (almost) always worked from home and that hasn't changed; (almost) all hours from home due to the pandemic; partially from home/partially at my normal workplace due to the pandemic; (almost) all hours at my normal workplace with the possibility to work from home; at my normal workplace because my work cannot be done from home; at home, but temporarily out of work due to pandemic; not applicable. We recoded this variable into (0) working from home; (1) working partially from home; (2) working at workplace with the possibility to work at home; (3) working at workplace due to the nature of the work. Respondents who were temporarily out of work and those who indicated not applicable were coded as missing. This ranged between 8 and 28 respondents depending on the wave (See S3 Table).

The relative division of childcare was measured on a 7-point Likert scale using the following question: "How do you and your partner/spouse divide the care for your child(ren) **right now** (including home schooling/help with homework)?" Answer categories ranged from (1) I do almost everything to (7) my partner/spouse does almost everything. These data were compared with retrospective pre-pandemic data and calculated as a change score, indicating the respondent doing relatively (1) more, (2) the same, or (3) less childcare relative to their partner, compared to before the pandemic. Respondents without co-resident minor children were not included in these analyses. In addition, between 3 and 32 respondents with co-resident minor children did not answer this question and were therefore excluded from the analysis (see S3 Table).

Work-life balance was measured with the COVID-adapted Eurofound item: "How easy or difficult is it for you to combine your paid work with your caregiving responsibilities (including home schooling/help with homework) **since the general closure of schools and childcare centres**?" Answers were on a 5-point Likert scale, ranging from (1) very easy to (5) very difficult. We recoded this into (1) (very) easy, (2) not easy/not difficult, (3) (very) difficult. At each subsequent wave, respondents were asked: "How easy or difficult was it for you to combine

your paid work with care and support for people around you **in the last month**?" Those with missing values or who reported 'not applicable' were coded as missing, this ranged between 4 and 91 respondents per wave (see S3 Table). For a full overview of descriptive statistics of dependent variables per wave, see S4 Table.

**Independent variable.** Our independent variable was gender (male = 1, female = 0); the LISS panel does not include non-binary options. Descriptive statistics of the sample by gender and wave are reported in S2 Table.

**Covariates.** We included six covariates that had the potential to affect the association between gender and our outcome variables ([see 51]; for a discussion on the importance of including only theoretically-informed covariates): essential occupation of the respondent and the partner, work location of the partner, age and education of the respondent, the presence of co-resident minor children in the household, age of the youngest child, and work location autonomy.

Essential occupation of the respondent and their partner was included to control for the potential effect of occupational differences on the association between gender and our outcome variables because essential workers were given special dispensation (e.g., access to emergency childcare) but also faced potentially more difficult working conditions (e.g., increased working hours, greater risk of infection) at different stages throughout the pandemic [e.g., 52]. This covariate was measured by providing respondents with a government-developed list of essential occupations: "The government has indicated a number of occupations as 'essential occupations'. This included care (including youth care and social support), childcare, public transport, the food chain (e.g., supermarkets), transport industry, waste/garbage collection and processing, media and communication, education, emergency services, necessary government processes." Respondents were asked to indicate whether they and/or their partner worked in an essential occupation. Answer categories were yes or no (reference category). Waves 5 and 6 did not include questions about essential occupation. Values for wave 5 were imputed using values from wave 4. At the time of data collection for wave 6 (April 2022), there was no longer a governmental distinction of essential and non-essential occupations, therefore wave 6 does not include information on essential occupations. Next, we included the work location of the respondent's partner, coded in the same way as for respondents: (0) working from home; working partially from home (1); working at workplace with the possibility to work at home (2); working at workplace due to the nature of the work (3). Third, we included age as a covariate (in years) to account for potential differences in work and care combinations across the life course [53]. Our fourth covariate was education, as the relationship between gender and paid work, the relative division of childcare, and work-life balance can also be affected by differences in education [e.g., 54–57]. This variable was categorized into three levels (1) primary or secondary qualifications; (2) vocational qualifications (reference category); (3) tertiary education. The presence of (young) children has also been found to affect the relationship between gender and paid work, care work, and wellbeing [e.g., 58,59]. The presence of co-resident minor children in the household was measured as (1) yes or (0) no (reference category). We also included the age of the youngest child in our analyses of the relative division of childcare, which focused only on parents with co-resident minor children. Lastly, we controlled for potential differences in work autonomy (i.e., the degree to which employees or employers determine where work is carried out) as men and women may differ in the degree of autonomy available [60]. Work location autonomy was only measured in waves 2–6, using the following question: "I can decide where I work", with response categories of (0) disagree, (1) neutral (reference category), (2) agree, and (3) not applicable.

**Moderators.** Three of the covariates had the potential to moderate gender inequalities in work location, the relative division of childcare, and work-life balance: parenthood, working

in an essential occupation, and education. Men and women and employees with and without children differed in their reasons for working from home and their ability to do so prior to the pandemic [61–63], therefore parenthood is measured as a potential moderator. Second, we explored the idea that men and women in and outside of essential occupations differed in their work location [see also 63], the relative division of childcare, and work-life balance. Third, the relationship between gender and paid work, the relative division of childcare, and wellbeing is often socially stratified along educational lines [e.g., 54–57]. We therefore included education as a moderator.

**Data analysis.** Analyses proceeded in four steps. First, all outcome measures were measured descriptively. Second, we ran parsimonious, cross-sectional, multinomial logit models analysing gender in relation to each outcome measure in each wave. Although we had longitudinal data, running repeated cross-sections allowed for a comparison across waves, testing whether the effect of gender on a given outcome variable changed throughout the pandemic. In a third step, relevant covariates were added to the multinomial logit model. Results are presented as Average Marginal Effects (AMEs) to ease interpretation, allowing for comparisons across waves [64]. As a fourth step, we ran interaction models for each of our three moderators, presented as Marginal Effects at Representative values (MERs). We estimated the MERs for gender*parenthood (men without children compared to women without children; fathers compared to mothers), for gender*essential occupation (comparing men and women without an essential occupation; and men and women with an essential occupation), and for gender*education (men compared to women across the different educational groups).

Full moderation models for each wave including the MERs can be found in the supplemental files (see S6–S8 Tables for work location, S9 and S10 Tables for relative division of childcare, and S11–S13 Tables for work-life balance); only the most relevant findings were discussed below. Note that interaction models were limited to those respondents with relevant information: as the first wave only included parents, we could not estimate interaction effects for gender and having co-resident minor children in April 2020. The models with the relative division of childcare as the dependent variable focused only on parents. Furthermore, essential occupation and essential occupation of the partner were not measured in November 2021 (wave 5) or April 2022 (wave 6). For wave 5, we imputed these variables using data from previous waves (wave 4 if available, otherwise wave 3, otherwise wave 2, and otherwise wave 1). We did not impute essential occupation or essential occupation of the partner for wave 6 as all containment measures had been lifted, thus no distinction was in effect between essential and non-essential occupations. We therefore did not estimate any interaction effects with essential occupation for wave 6.

Finally, we conducted a robustness check on the sub-sample of parents for the outcome work-life balance in order to include the relative division of childcare as a covariate. In these analyses, missing values were deleted for childcare and age youngest child. Therefore, the N for each wave is slightly lower compared to the main analyses.

## Results

### A description of the gender gap at the start of the pandemic

Amongst our respondents, hardly any gender difference is visible in who worked solely from home prior to the pandemic, our only pre-pandemic measure of working from home, with 8.2 per cent of men and 9.1 per cent of women reporting this work pattern (see S5 Table). We note, however, that overall, working from home was uncommon in the Netherlands prior to the pandemic, with 3 out of 10 employees occasionally working from home [65].

The division of childcare was unequally divided between men and women prior to the pandemic. More than half (60%) of mothers reported doing (much) more childcare compared to their partner, compared to 4 per cent of fathers (see S5 Table). Just more than a third (37%) of mothers and fathers (35%) reported an equal division of childcare pre-pandemic. Fathers reported the highest percentage of doing (much) less childcare than their partner pre-pandemic (61% versus 4% of mothers).

For work-life balance, most respondents' pre-pandemic perception of combining paid work with care was either that this was not very difficult or was neither easy nor difficult (see S5 Table). However, some gender differences are apparent. Around half of women (48%) reported that work-life balance was (very) easy prior to the pandemic compared to 58 per cent of men, a difference of ten percentage points. More women (38%) than men (31%) reported it was neither easy nor difficult to combine work and care prior to the pandemic.

In short, prior to the pandemic, some gender differences were evident in work location and perceived work-life balance. The relative division of care was clearly unequal, with mothers reporting a much higher share of care than fathers pre-pandemic.

## Phase-specific gender differences throughout the pandemic

In the multivariate analyses controlling for covariates, we find several consistent patterns across each wave of differences between men and women in work location, the relative division of childcare, and work-life balance. Most of the gender differences found changed throughout the pandemic, with the findings from our repeated cross-sectional analyses suggesting that certain gender differences are phase specific.

Gender differences in work location (see S14–S19 Tables) show that men were less likely than women to work fully from home at the start of each lockdown (April 2020, November 2020, and November 2021). This difference declined gradually, from 7 (percentage points) p.p. difference at the start of the first lockdown, to 6 p.p. at the second lockdown and 5 p.p. at the third lockdown. Moreover, men were more likely than women to work hybrid (partially from home, partially on location) but only during the first lockdown (April 2020 and June 2020, with 5 p.p. and 6 p.p. difference respectively). Men were also more likely than women to work on location despite having the possibility to work from home during the first lockdown (April 2020 and September 2020, with 4 p.p. and 6 p.p. difference respectively). The findings on working on location due to the nature of the work are mostly insignificant. We find only that men were 5 p.p. more likely than women to work on location due to the nature of the work at the start of the third lockdown (November 2021).

In relation to gender differences in childcare (see S20–S25 Tables), we see that at different phases in the pandemic, fathers reported doing more childcare than before. Fathers were between 8 and 9 p.p. more likely than mothers to report doing more childcare than prior to the pandemic throughout the first phase of the pandemic (9 p.p. in April and June 2020; 8 p.p. in September 2020, see S20–S22 Tables). Initially, the results suggested that the end of the first phase of the pandemic was a turning point for fathers, with fathers being more likely to report doing the same amount of childcare as before the pandemic between September 2020 and November 2021 (10 p.p., 15 p.p. and 10 p.p. respectively; S22–S25 Tables). At the start of the second and third lockdowns (November 2020 and November 2021), fathers are no longer significantly more likely than mothers to report doing more childcare than prior to the pandemic. However, at the end of pandemic (April 2022), we see that some fathers are more likely than mothers to report doing more childcare than prior to the pandemic (10 p.p.).

Some fathers were also less likely than mothers to report doing *less* childcare than prior to the pandemic. In the middle of the first phase (June 2020), fathers were 9 p.p. less likely than

mothers to report doing less childcare and this remained consistent throughout the pandemic. This gender difference initially increased (to 18 p.p. in September 2020; 14 p.p. in November 2020) before returning to around ten percentage points (9 p.p. at the start of the third lockdown, November 2021; 12 p.p. at the end of the pandemic, April 2022).

Few discernible patterns are visible in relation to gender differences in work-life balance (S26–**S31** Tables). Men were more likely than women to state work-life balance was easy in the middle of the first lockdown (June 2020; 12 p.p.) and again at the start of the second lockdown (November 2020; 9 p.p.). In contrast, women were more likely to report finding work-life balance neither easy nor difficult during the middle of the first lockdown phase (10 p.p., June 2020).

## Drivers of pandemic-based gender inequality

We tested three potential moderators of the relationship between gender and the outcomes of work location and work-life balance: parenthood, having an essential occupation, and education. For the relative division of childcare, we tested having an essential occupation and education as potential moderators. The full results of these interaction models are presented in S6–S13 Tables.

In relation to work location, gender differences appear to primarily be driven by working in an essential occupation. Amongst workers without an essential occupation, men were significantly less likely (11 p.p.) than women to work fully from home (see Fig 1). This difference remained throughout the pandemic, except for September 2020, when many workplaces (temporarily) re-opened. We also see that the gender difference in who worked on location due to the nature of the work was moderated by having an essential occupation (see Fig 2). By the middle of the first lockdown, men with an essential occupation were 13 p.p. less likely than women with an essential occupation to work on location due to the nature of the work.

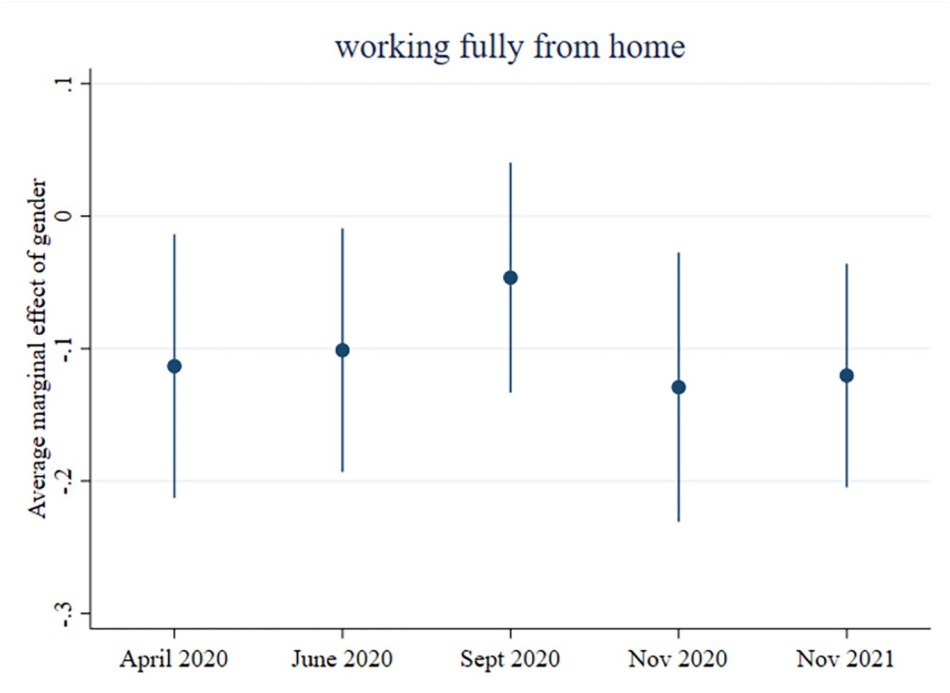

**Fig 1. Men working fully from home compared to women, in non-essential occupations.**

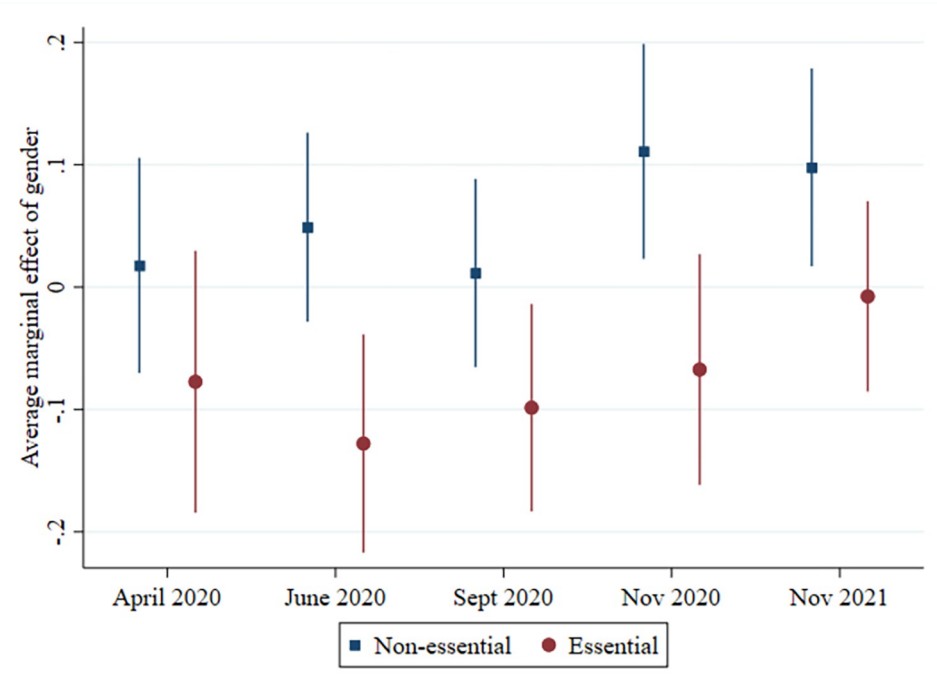

**Fig 2. Men working on location due to the nature of the work compared to women, by essential occupation.**

Education also appears to be driving some of the gender differences in work location. Men with primary or secondary education are less likely to work fully from home and more likely to work hybrid compared to women with primary or secondary education at various stages throughout the pandemic. These same men are also more likely to work on location due to the nature of the work (see S8 Table). Gender differences in work location are largely unrelated to whether or not respondents had co-resident minor children (see S6 Table).

In relation to the relative division of childcare, we also see that gender differences are moderated by having an essential occupation or not (see S9 Table). Compared to mothers in essential occupations, fathers in essential occupations were significantly less likely to report doing less childcare from the middle of the first lockdown throughout the third and final lockdown (see Fig 3). Fathers not working in an essential occupation were also less likely to report doing less childcare compared to mothers not working in an essential occupation at the beginning of each lockdown. Gender differences in who did more or less in the relative division of childcare throughout the pandemic are also moderated by education (see S10 Table). Although fathers without tertiary education initially did more childcare at the start of the first lockdown than mothers without tertiary education (11 p.p.), this finding was no longer significant by June 2020. Indeed, by the second and third lockdowns, fathers without tertiary education were more likely than mothers without tertiary education to report doing the same amount of childcare as prior to the pandemic. For vocationally educated fathers in particular, this result remained significant until the end of the pandemic in April 2022. They were more likely than mothers with vocational education to report doing the same amount of childcare as before. In contrast, fathers with a tertiary level of education were more likely than tertiary educated mothers to report doing more childcare compared to before the pandemic in the later phase of the pandemic. The decline in parents' relative share of childcare is also moderated by education: Fathers without tertiary education are less likely than mothers without tertiary education

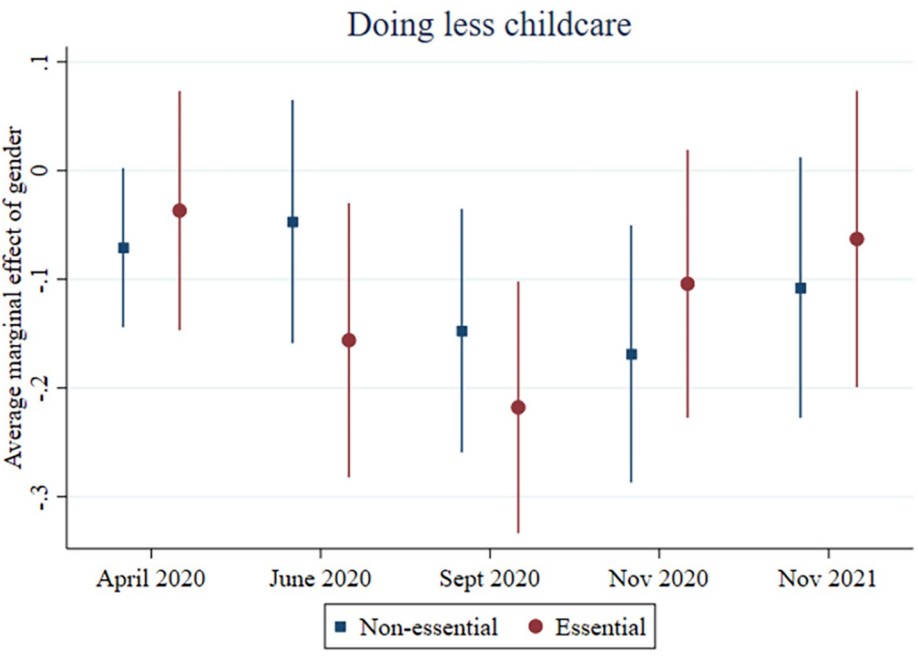

**Fig 3. Fathers doing less childcare (compared to mothers), by essential occupation.**

to report doing less childcare by the end of the first lockdown, an association which remains for vocationally educated fathers throughout the second and third lockdowns (see Fig 4 and S10 Table).

In line with the absence of significant effects in the multivariate models without moderators, few discernible patterns are visible in relation to gender differences in work-life balance in relation to parenthood (see S11 Table), having an essential occupation (see S12 Table) or education (see S13 Table). The gender difference found in the middle of the first lockdown and the start of the second lockdown is primarily related to parenthood. In the middle of the first lockdown, men with (11 p.p.) and without co-resident minor children (17 p.p.) were both more likely to report that work-life balance was easy compared to women with and without co-resident minor children respectively. By the start of the second lockdown, only men without co-resident minor children found it easier (19 p.p.) than women without co-resident minor children to combine work and care. Additionally, men with co-resident minor children were less likely to say that work-life balance was difficult at the start of the second lockdown (4 p.p.) compared to women with co-resident minor children and again by the end of the pandemic (5 p.p.).

The findings from our robustness checks confirmed the role parenthood plays in the association between gender and work-life balance. The analysis of perceived work-life balance for the sub-sample of respondents with co-resident minor children suggests that amongst parents, gender differences were evident in who found it easy or difficult to combine work and care at various phases throughout the pandemic (see S32–S37 Tables). Fathers were between 6–7 p.p. less likely than mothers to report finding the combination of work and care difficult at the start of the first and second lockdowns. By the end of the pandemic, fathers were 9 p.p. less likely than mothers to report difficulty combining work and care. Fathers were also more likely than mothers to report that combining work and care was easy in the middle of the first lockdown (13 p.p.) and again by the time the pandemic ended (April 2022; 11 p.p.).

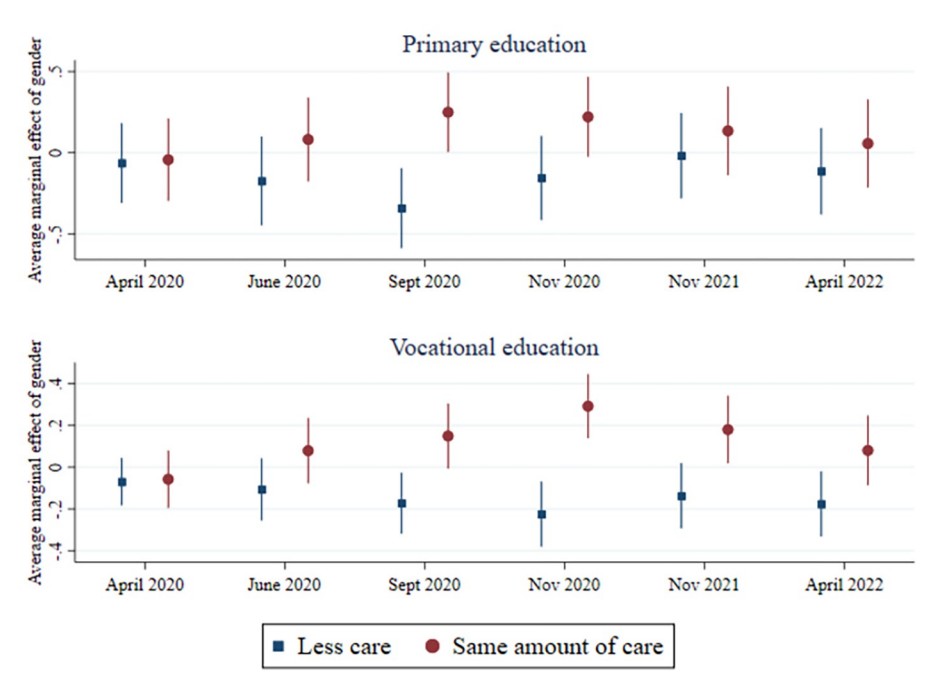

**Fig 4. Fathers without tertiary education doing less or same amount of childcare (compared to mothers without tertiary education).**

## Discussion

Given concerns at the start of the pandemic that government containment measures to reduce the spread of the Sars-Cov2 virus would impact women and men differently, thereby exacerbating existing inequalities in paid work [1,2], unpaid work (e.g., childcare) [3], and wellbeing [4], an investigation into the development of potential differences across multiple stages of the pandemic is warranted. A key finding of this study is that the COVID-19 pandemic is not associated with a unilateral worsening of existing inequalities between men and women in the Netherlands. We also find evidence that differences between men and women were often phase-specific, suggesting gendered patterns of sensemaking throughout the pandemic.

In relation to work location, despite general government measures requiring employees to work from home (both in the spring of 2020 and again in the fall/winter of 2021; see timeline in S1 Fig), men were less likely than women to work fully from home at the start of each lockdown. Moreover, as offices reopened, men were more likely than women to work on location even if the work did not require them to do so, whereas women were more likely than men to have to work on location due to the nature of the work. These findings confirm studies on the initial phases of the pandemic that women were more likely than men to be affected by pandemic-induced changes to paid work [9,10,63]. They also suggest that men's sensemaking revolved more around maintaining a presence at the workplace. Women's sensemaking, in contrast, appears to have been more centred on working fully from home unless they were working in essential occupations, much of which had to be done on-site. These findings raise concerns about the longer-term impact of these developments [e.g., 66]. For example, women potentially faced higher risks of infection because they more often had to work on location due to the nature of the work [e.g., 67]. At the same time, men who chose to work on location could benefit from improved career development, if they are perceived to have greater work

commitment than women [68]. From a policy perspective, the findings point to potential varied interpretations of and/or adherence to government containment measures [69,70], which could be accounted for when developing containment measures for future pandemics.

In the relative division of childcare, our finding that the pandemic differentially affected men and women in relation to divisions of childcare continues to nuance the growing evidence base on this topic [13–15,18–20]. The relative division of childcare between mothers and fathers was and remains unequal in the Netherlands [71,72]. However, two years after the start of the pandemic, a proportion of fathers continues to report doing relatively more childcare than before, a finding which is in line with US research on the first year of the pandemic [16] but contrasts findings in other countries, like Germany, where fathers' increase in relative childcare was temporary [73]. Mothers also remain more likely than fathers to report doing less childcare than prior to the pandemic. We cannot rule out that these findings reflect the differential starting point for mothers and fathers, which makes it more likely that fathers will move upwards and mothers downwards in their relative division of childcare (e.g., a plateau or ceiling effect). However, it could be that our findings reflect gendered patterns of sensemaking in the Netherlands, as identity is closely linked to sensemaking processes [74]. Government containment measures impacted parents greatly, with the closure of childcare centres and schools requiring parents to increase the time spent on childcare and schooling when working from home [75,76]. Gendered sensemaking could be reflected in how some fathers responded to containment measures, in particular working from home mandates, as these appear to have created opportunities to notice the need for childcare and to increase their involvement in it [e.g., 20]. However, fathers in some countries returned to their pre-pandemic gendered divisions of childcare as the pandemic continued [e.g., 73]. Within the Netherlands, differences were also found within groups of parents. Tertiary educated mothers and fathers are more likely to share childcare relatively equally than parents without tertiary education [77]. The finding that the interaction between gender and education in the relative division of childcare was driven by differences between primary/secondary educated mothers and fathers, and later by differences between tertiary educated mothers and fathers could suggest that changes in the relative division of childcare primarily strengthen a process that was already occurring. Namely, the group still doing more childcare than prior to the pandemic by April 2022 (tertiary educated fathers compared to tertiary educated mothers), was already more likely to have a relatively equal division of childcare pre-pandemic [72,78]. From a sensemaking perspective, this finding fits with underlying traditional gender roles in the Netherlands in which mothers are seen as better caregivers than fathers, with men taking on a primary breadwinner role [72,77,79]. Investing in these roles as mother/carer and father/provider might have increased sensemaking for non-tertiary educated mothers and fathers, especially during the lockdowns.

The relative absence of differences between men and women in the perceived ease or difficulty in combining work and care throughout the pandemic is notable. Studies on the initial stages of the pandemic suggest that overall wellbeing declined, particularly for mothers [e.g., 21,22]. In relation to work-life balance, however, which is one aspect of wellbeing, our findings demonstrated only nuanced differences, with men at times more likely to report finding it (very) easy to combine work and care and women more likely to report finding it neither easy or difficult. These subtle differences could be a reflection of construct bias, insofar that women and men may have differing understandings of 'easy' in relation to combining work and care. On the other hand, these findings might support the idea that experiences of wellbeing in general and work-life balance in particular often differ across countries [80,81]. Our robustness check on the sub-sample of parents does suggest that within this group, however, important gender differences exist. Compared to mothers, fathers reported fewer difficulties combining work and care at the start of the first and second lockdowns and by the end of the pandemic.

Fathers also reported greater ease than mothers in combining work and care in the middle of the first lockdown and at the end of the pandemic. The findings from the robustness check are therefore in line with existing studies on wellbeing, that show a larger decline in wellbeing particularly for mothers [e.g., 21,22].

## Limitations and future research

We note a number of methodological limitations. The analyses provided here are based on self-reported measures as well as several retrospective measures, which could potentially introduce bias. Given the relatively low number of missing values across variables, we chose for listwise deletion of cases rather than a multiple imputation strategy, which also could introduce bias. However, all missing data strategies have advantages and disadvantages, and alternatives, such as multiple imputation, can lead to the overestimation of effects [82]. Our study is also limited in the generalizability of findings given the focus on a single country, and the study does not address issues of causality. However, by providing descriptive longitudinal and repeated cross-sectional evidence on the development of gender inequalities throughout two years of the pandemic using representative, probability-based data, we are able to extend existing pandemic findings often limited to the first months of the pandemic. Additionally, our study was exploratory in its analysis of the drivers of gender differences in work location, the relative division of childcare, and work-life balance, focusing on interactions between gender and parenthood, working in an essential occupation, and education to understand gendered patterns of sensemaking across various phases of the pandemic. Analyses focused on unpacking any of these relationships in greater detail could consider other potential confounders, such as public/private sector differences in work location, or differences between the absolute and relative division of childcare. We also did not control for the health of individuals, another potential confounder, which is, for example, associated with work-life balance [83].

Limitations aside, our study offers important inroads for future research into the long-term societal impact of the COVID-19 pandemic. The gender differences in work location suggest ongoing attention is needed for longer-term changes to work patterns. As more countries declare an end to the COVID-19 pandemic, fundamental changes to how work is organized, particularly possibilities to work from home, have the potential to deepen gender inequalities in paid work if insufficient attention is given to differences in men and women's work locations [20,75]. In relation to the relative division of childcare, future research would do well to focus on qualitative evidence [e.g., 84], for example on fathers' experiences of increased childcare and mothers' experiences of doing relatively less childcare. Such research would allow greater theorizing on the mechanisms that could potentially lead to more equal divisions of childcare in the longer-term. Longitudinal research is also needed to follow the developments in childcare divisions following the pandemic [16,73]. Because overall, gender differences in the relative division of childcare did not worsen but declined at various stages in the Netherlands. These pandemic-based changes could lead to longer-term changes in the relative division of childcare, even though mothers continue to do more than fathers. Finally, our findings also offer a starting point for future comparative research, for example to understand why the Dutch case shows a potential medium-term improvement in gendered divisions of childcare compared to countries like Germany, where such improvements have not occurred [73].

## Conclusion

Our conclusions are threefold. First, differences between men and women are not unequivocally evident in all domains during the pandemic: Our findings suggest gender differences were primarily found in work location and the relative division of childcare. In relation to

work-life balance, differences between men and women in the extent to which they found it easy or difficult to combine paid work with care responsibilities were nuanced and inconsistent throughout the pandemic, and primarily visible between mothers and fathers. Second, the analyses show that many gender differences are phase-specific, suggesting potential differences in how women and men made sense of the upheaval of the pandemic in relation to work and care. Third, essential occupation and education were both drivers of many of these phase-specific gender differences. Taken together, these findings suggest that inequalities between men and women in paid work, unpaid work (e.g., childcare), and wellbeing have neither been alleviated by nor unilaterally worsened during the COVID-19 pandemic and that attention is needed for the drivers of these differences as well their phase-specific nature.

## Supporting information

**S1 Fig. Timeline of data collection in relation to relevant pandemic measures in the Netherlands.**
(PDF)

**S1 Table. Excluded cases by wave based on sample selection and missing values (N).** N/A: variable not measured in that wave; 0 = no missings.
(DOCX)

**S2 Table. Descriptive statistics by wave.**
(DOCX)

**S3 Table. Excluded cases by wave (N)–dependent variables.**
(DOCX)

**S4 Table. Descriptive statistics of dependent variables (work location, division of childcare and work-life balance), by wave.**
(DOCX)

**S5 Table. Descriptive statistics of dependent variables (work location, division of childcare and work-life balance) prior to the pandemic, by gender.** Note: Work location and work-life balance based on first wave that included both people with and without children (wave 2), childcare based on wave 1. For pre-pandemic work-life balance there were 8 missing values.
(DOCX)

**S6 Table. Marginal effect of gender on work location, with and without (w/o) minor, co-resident children.** Note: Standard errors in parentheses. *** p<0.01, ** p<0.05, * p<0.1. Controlled for all co-variates. Reference categories are women, non-essential occupations, partner in non-essential occupation, vocational education, no minor co-resident children, neutral on statement 'I can decide where I work', partner working on location due to the nature of the work.
(DOCX)

**S7 Table. Marginal effect of gender on work location in essential and non-essential occupations.** Note: Standard errors in parentheses. *** p<0.01, ** p<0.05, * p<0.1. Controlled for all co-variates. Reference categories are women, non-essential occupations, partner in non-essential occupation, vocational education, no minor co-resident children, neutral on statement 'I can decide where I work', partner working on location due to the nature of the work.
(DOCX)

**S8 Table. Marginal effect of gender on work location across educational groups.** Note: Standard errors in parentheses. *** p<0.01, ** p<0.05, * p<0.1. Controlled for all co-variates.

Reference categories are women, non-essential occupations, partner in non-essential occupation, vocational education, no minor co-resident children, neutral on statement 'I can decide where I work', partner working on location due to the nature of the work.
(DOCX)

**S9 Table. Marginal effect of gender on division of childcare tasks in essential and non-essential occupations.** Note: Standard errors in parentheses. *** p<0.01, ** p<0.05, * p<0.1. Controlled for all co-variates. Reference categories are women, non-essential occupations, partner in non-essential occupation, vocational education, no minor co-resident children, neutral on statement 'I can decide where I work', partner working on location due to the nature of the work. Essential occupation was not measured in wave 6 (April 2022) and is therefore excluded from these analyses.
(DOCX)

**S10 Table. Marginal effect of gender on division of childcare tasks across educational groups.** Note: Standard errors in parentheses. *** p<0.01, ** p<0.05, * p<0.1. Controlled for all co-variates. Reference categories are mothers, non-essential occupations, partner in non-essential occupation, vocational education, no minor co-resident children, neutral on statement 'I can decide where I work', partner working on location due to the nature of the work.
(DOCX)

**S11 Table. Marginal effect of gender on work-life balance, with and without (w/o) minor co-resident children.** Note: Standard errors in parentheses. *** p<0.01, ** p<0.05, * p<0.1. Controlled for all co-variates. Reference categories are women, non-essential occupations, partner in non-essential occupation, vocational education, no minor co-resident children, neutral on statement 'I can decide where I work', partner working on location due to the nature of the work.
(DOCX)

**S12 Table. Marginal effect of gender on work-life balance in essential and non-essential occupations.** Note: Standard errors in parentheses. *** p<0.01, ** p<0.05, * p<0.1. Controlled for all co-variates. Reference categories are women, non-essential occupations, partner in non-essential occupation, vocational education, no minor co-resident children, neutral on statement 'I can decide where I work', partner working on location due to the nature of the work.
(DOCX)

**S13 Table. Marginal effect of gender on work-life balance across educational groups.** Note: Standard errors in parentheses. *** p<0.01, ** p<0.05, * p<0.1. Controlled for all co-variates. Reference categories are women, non-essential occupations, partner in non-essential occupation, vocational education, no minor co-resident children, neutral on statement 'I can decide where I work', partner working on location due to the nature of the work.
(DOCX)

**S14 Table. Multinomial logits of work location, including estimated average marginal effects of all covariates in April 2020.** Note: *** p<0.01, ** p<0.05, * p<0.1. Reference categories are women, non-essential occupations, partner in non-essential occupation, vocational education, partner working on location due to the nature of the work.
(DOCX)

**S15 Table. Multinomial logits of work location, including estimated average marginal effects of all covariates in June 2020.** Note: *** p<0.01, ** p<0.05, * p<0.1. Reference

categories are women, non-essential occupations, partner in non-essential occupation, vocational education, no minor co-resident children, neutral on statement 'I can decide where I work', partner working on location due to the nature of the work.
(DOCX)

**S16 Table. Multinomial logits of work location, including estimated average marginal effects of all covariates in September 2020.** Note: *** $p<0.01$, ** $p<0.05$, * $p<0.1$. Reference categories are women, non-essential occupations, partner in non-essential occupation, vocational education, no minor co-resident children, neutral on statement 'I can decide where I work', partner working on location due to the nature of the work.
(DOCX)

**S17 Table. Multinomial logits of work location, including estimated average marginal effects of all covariates in November 2020.** Note: *** $p<0.01$, ** $p<0.05$, * $p<0.1$. Reference categories are women, non-essential occupations, partner in non-essential occupation, vocational education, no minor co-resident children, neutral on statement 'I can decide where I work', partner working on location due to the nature of the work.
(DOCX)

**S18 Table. Multinomial logits of work location, including estimated average marginal effects of all covariates in November 2021.** Note: *** $p<0.01$, ** $p<0.05$, * $p<0.1$. Reference categories are women, non-essential occupations, partner in non-essential occupation, vocational education, no minor co-resident children, neutral on statement 'I can decide where I work', partner working on location due to the nature of the work.
(DOCX)

**S19 Table. Multinomial logits of work location, including estimated average marginal effects of all covariates in April 2022.** Note: *** $p<0.01$, ** $p<0.05$, * $p<0.1$. Reference categories are women, non-essential occupations, partner in non-essential occupation, vocational education, no minor co-resident children, neutral on statement 'I can decide where I work', partner working on location due to the nature of the work.
(DOCX)

**S20 Table. Multinomial logits of division of childcare, including estimated average marginal effects of all covariates in April 2020.** Note: *** $p<0.01$, ** $p<0.05$, * $p<0.1$. Reference categories are mothers, non-essential occupations, partner in non-essential occupation, vocational education, partner working on location due to the nature of the work.
(DOCX)

**S21 Table. Multinomial logits of division of childcare, including estimated average marginal effects of all covariates in June 2020.** Note: *** $p<0.01$, ** $p<0.05$, * $p<0.1$. Reference categories are mothers, non-essential occupations, partner in non-essential occupation, vocational education, neutral on statement 'I can decide where I work', partner working on location due to the nature of the work.
(DOCX)

**S22 Table. Multinomial logits of division of childcare, including estimated average marginal effects of all covariates in September 2020.** Note: *** $p<0.01$, ** $p<0.05$, * $p<0.1$. Reference categories are mothers, non-essential occupations, partner in non-essential occupation, vocational education, neutral on statement 'I can decide where I work', partner working on location due to the nature of the work.
(DOCX)

**S23 Table. Multinomial logits of division of childcare, including estimated average marginal effects of all covariates in November 2020.** Note: *** $p<0.01$, ** $p<0.05$, * $p<0.1$. Reference categories are mothers, non-essential occupations, partner in non-essential occupation, vocational education, neutral on statement 'I can decide where I work', partner working on location due to the nature of the work.
(DOCX)

**S24 Table. Multinomial logits of division of childcare, including estimated average marginal effects of all covariates in November 2021.** Note: *** $p<0.01$, ** $p<0.05$, * $p<0.1$. Reference categories are mothers, non-essential occupations, partner in non-essential occupation, vocational education, neutral on statement 'I can decide where I work', partner working on location due to the nature of the work.
(DOCX)

**S25 Table. Multinomial logits of division of childcare, including estimated average marginal effects of all covariates in April 2022.** Note: *** $p<0.01$, ** $p<0.05$, * $p<0.1$. Reference categories are mothers, non-essential occupations, partner in non-essential occupation, vocational education, neutral on statement 'I can decide where I work', partner working on location due to the nature of the work.
(DOCX)

**S26 Table. Multinomial logits of work-life balance, including estimated average marginal effects of all covariates in April 2020.** Note: *** $p<0.01$, ** $p<0.05$, * $p<0.1$. Reference categories are women, non-essential occupations, partner in non-essential occupation, vocational education, partner working on location due to the nature of the work.
(DOCX)

**S27 Table. Multinomial logits of work-life balance, including estimated average marginal effects of all covariates in June 2020.** Note: *** $p<0.01$, ** $p<0.05$, * $p<0.1$. Reference categories are women, non-essential occupations, partner in non-essential occupation, vocational education, no minor co-resident children, neutral on statement 'I can decide where I work', partner working on location due to the nature of the work.
(DOCX)

**S28 Table. Multinomial logits of work-life balance, including estimated average marginal effects of all covariates in September 2020.** Note: *** $p<0.01$, ** $p<0.05$, * $p<0.1$. Reference categories are women, non-essential occupations, partner in non-essential occupation, vocational education, no minor co-resident children, neutral on statement 'I can decide where I work', partner working on location due to the nature of the work.
(DOCX)

**S29 Table. Multinomial logits of work-life balance, including estimated average marginal effects of all covariates in November 2020.** Note: *** $p<0.01$, ** $p<0.05$, * $p<0.1$. Reference categories are women, non-essential occupations, partner in non-essential occupation, vocational education, no minor co-resident children, neutral on statement 'I can decide where I work', partner working on location due to the nature of the work.
(DOCX)

**S30 Table. Multinomial logits of work-life balance, including estimated average marginal effects of all covariates in November 2021.** Note: *** $p<0.01$, ** $p<0.05$, * $p<0.1$. Reference categories are women, non-essential occupations, partner in non-essential occupation, vocational education, no minor co-resident children, neutral on statement 'I can decide where I

work', partner working on location due to the nature of the work.
(DOCX)

**S31 Table. Multinomial logits of work-life balance, including estimated average marginal effects of all covariates in April 2022.** Note: *** p<0.01, ** p<0.05, * p<0.1. Reference categories are women, non-essential occupations, partner in non-essential occupation, vocational education, no minor co-resident children, neutral on statement 'I can decide where I work', partner working on location due to the nature of the work.
(DOCX)

**S32 Table. Robustness check: Multinomial logits of work-life balance, including estimated average marginal effects of all covariates in April 2020, sub-sample of parents with co-resident minor children.** Note: *** p<0.01, ** p<0.05, * p<0.1. Reference categories are mothers, non-essential occupations, spouse in non-essential occupation, vocational education, partner works on location by nature of work, less childcare.
(DOCX)

**S33 Table. Robustness check: Multinomial logits of work-life balance, including estimated average marginal effects of all covariates in June 2020, sub-sample of parents with co-resident minor children.** Note: *** p<0.01, ** p<0.05, * p<0.1. Reference categories are mothers, non-essential occupations, spouse in non-essential occupation, vocational education, neutral on workplace autonomy, partner works on location by nature of work, less childcare.
(DOCX)

**S34 Table. Robustness check: Multinomial logits of work-life balance, including estimated average marginal effects of all covariates in September 2020, sub-sample of parents with co-resident minor children.** Note: *** p<0.01, ** p<0.05, * p<0.1. Reference categories are mothers, non-essential occupations, spouse in non-essential occupation, vocational education, neutral on workplace autonomy, partner works on location by nature of work, less childcare.
(DOCX)

**S35 Table. Robustness check: Multinomial logits of work-life balance, including estimated average marginal effects of all covariates in November 2020, sub-sample of parents with co-resident minor children.** Note: *** p<0.01, ** p<0.05, * p<0.1. Reference categories are mothers, non-essential occupations, spouse in non-essential occupation, vocational education, neutral on workplace autonomy, partner works on location by nature of work, less childcare.
(DOCX)

**S36 Table. Robustness check: Multinomial logits of work-life balance, including estimated average marginal effects of all covariates in November 2021, sub-sample of parents with co-resident minor children.** Note: *** p<0.01, ** p<0.05, * p<0.1. Reference categories are mothers, non-essential occupations, spouse in non-essential occupation, vocational education, neutral on workplace autonomy, partner works on location by nature of work, less childcare.
(DOCX)

**S37 Table. Robustness check: Multinomial logits of work-life balance, including estimated average marginal effects of all covariates in April 2022, sub-sample of parents with co-resident minor children.** Note: *** p<0.01, ** p<0.05, * p<0.1. Reference categories are mothers, vocational education, neutral on workplace autonomy, partner works on location by nature of work.
(DOCX)

## Acknowledgments

The authors would like to thank former CoGIS-NL project members Debby Beckers, Bryn Hummel, Sabine Geurts, and Peter Kruyen as well as two anonymous reviewers for their detailed and useful feedback on earlier versions of this manuscript.

## Author Contributions

**Conceptualization:** Mara A. Yerkes, Janna Besamusca, Roos van der Zwan, Stéfanie André, Chantal Remery, Ilse Peeters.

**Formal analysis:** Mara A. Yerkes, Janna Besamusca, Roos van der Zwan, Stéfanie André, Chantal Remery, Ilse Peeters.

**Funding acquisition:** Mara A. Yerkes, Janna Besamusca, Roos van der Zwan, Stéfanie André.

**Methodology:** Mara A. Yerkes, Janna Besamusca, Roos van der Zwan, Stéfanie André, Chantal Remery, Ilse Peeters.

**Project administration:** Mara A. Yerkes.

**Writing – original draft:** Mara A. Yerkes, Janna Besamusca, Roos van der Zwan, Stéfanie André, Chantal Remery.

**Writing – review & editing:** Mara A. Yerkes, Janna Besamusca, Roos van der Zwan, Stéfanie André, Chantal Remery, Ilse Peeters.

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
