## [Decision Letter · Decision Letter 0]

12 Sep 2023

PONE-D-23-16207Gender inequality in work, childcare and wellbeing: the longitudinal impact of the COVID-19 pandemicPLOS ONE

Dear Dr. Yerkes,

Thank you for submitting your manuscript to PLOS ONE. After careful consideration, we feel that it has merit but does not fully meet PLOS ONE’s publication criteria as it currently stands. Therefore, we invite you to submit a revised version of the manuscript that addresses the points raised during the review process. Please submit your revised manuscript by Oct 27 2023 11:59PM. If you will need more time than this to complete your revisions, please reply to this message or contact the journal office at plosone@plos.org. Please include the following items when submitting your revised manuscript:A rebuttal letter that responds to each point raised by the academic editor and reviewer(s). You should upload this letter as a separate file labeled 'Response to Reviewers'.A marked-up copy of your manuscript that highlights changes made to the original version. You should upload this as a separate file labeled 'Revised Manuscript with Track Changes'.An unmarked version of your revised paper without tracked changes. You should upload this as a separate file labeled 'Manuscript'.

We look forward to receiving your revised manuscript.

Kind regards,

Florian Fischer

Academic Editor

PLOS ONE

3. We notice that your supplementary figures are uploaded with the file type 'Figure'. Please amend the file type to 'Supporting Information'. Please ensure that each Supporting Information file has a legend listed in the manuscript after the references list.

Reviewers' comments:

Reviewer's Responses to Questions

**Comments to the Author**

1. Is the manuscript technically sound, and do the data support the conclusions?

Reviewer #1: Partly

Reviewer #2: Partly

2. Has the statistical analysis been performed appropriately and rigorously? 

Reviewer #1: N/A

Reviewer #2: N/A

3. Have the authors made all data underlying the findings in their manuscript fully available?

Reviewer #1: No

Reviewer #2: No

4. Is the manuscript presented in an intelligible fashion and written in standard English?

Reviewer #1: No

Reviewer #2: Yes

5. Review Comments to the Author

Reviewer #1: This study addresses the diachronic evolution of gender differences in work arrangements, childcare division, and work-life balance.

The descriptive findings extend the existing knowledge by covering approximately two years since the outbreak of the pandemic emergency. The topic is highly relevant for the literature addressing gender inequality and COVID-19-related inequalities, and the paper proposes to make a significant contribution to these readerships. However, major revisions are currently required to meet the publication criteria.

My major points are:

1) INTRODUCTION AND RESEARCH QUESTION: I personally appreciate the review of existing studies on the topic produced over the last few years, how the existing evidence is critically discussed, and how the question about the evolution of existing disparity over the 2020-2022 period emerges. However, the hypotheses appear poorly motivated and are built upon previous works, primarily from the UK. Although the article does not seem to engage in a detailed theoretical discussion, I believe that the authors should discuss the theoretical mechanisms behind the discussed expectations and the theoretical roles of the mentioned moderators. Furthermore, these motivated expectations require adequate contextualization given the country under scrutiny.

2) NOT CLEAR SAMPLE STRATEGY: I found it difficult to understand who composed the sample. On page 7, the authors specify that the survey was handed to LISS panel members with at least one working member and a co-resident minor child. However, in the analyses, having or not a child became a control variable (or parenthood as a moderator). Descriptive statistics for the variables of interest were missing. Moreover, if the sample can or cannot have children, the analysis of the childcare division necessarily has a sample different from that of the other analyses. This should be discussed and a description of who composes the different samples should be provided.

3) SELECTION IN SAMPLE: The sampling criteria and two outcome variables require the respondents to be in paid employment. As many of the COVID-19 implications, particularly on gender inequalities, have also occurred by affecting (un-)employment chances, you specifically make a selection of individuals that are still employed in each wave. This selection is not incorrect, but it should also be discussed in relation to the composition of this sample between men and women. Finally, is this selection too strong for the analysis of childcare divisions?

4) OUTCOME VARIABLES: Although I like the pre-post operationalization of the outcome variables (i.e. what is the difference with the pre-pandemic period), it would be anyway interesting to look at the absolute values of each wave. This is especially true for childcare, as children may grow over two or more years, and changes in the degree of childcare involvement may be driven by the aging of the child. Moreover, one may also consider the construction of a “within-couple disparity” measure for childcare, where the outcome becomes an imbalance (or changes in the balance) between the two partners.

5) CONTROL VARIABES: In the analyses, other theoretically driven covariates should be considered. The age of the (youngest) child, for instance, is indicative of the required childcare, the number of children also corrects for differential needs between families, and the biological sex of the child also matters for parent-child interaction. Moreover, the partners’ characteristics and health conditions should be considered. Finally, childcare involvement (and particularly paternal childcare involvement) should also be regarded as a control variable in the study of work-life balance (if all the sampled individuals have at least one child, but indeed this is not clear). For men, it may be easier to combine work and life duties if they do not carry-on childcare activities at all, so men with similar childcare burdens across time should be compared.

6) RESULTS: Here is the major point of the review. While going through the paper, I felt that there were simply too many aspects under consideration that hampered a smooth reading and understanding of the contribution. First, the three outcomes are only marginally related to each other, sometimes referring to different reference populations, and in the end, they appear to be chosen more for convenience than for specific theoretical reasons. Second, these outcomes are with multiple categories. Even though the multinomial approach is the correct one, the amount of information to be commented on for each analysis (also considering the two/three moderators) is overwhelming, and I took several readings to grasp them. Alternative operationalizations may help simplify the analyses and communication of the results. Third, the discussion of the results is currently not helpful in simplifying the provided information. In the current version, each result is discussed didactically, without any underlying narrative. Besides reconsidering the overly large focus of the contribution, I would suggest rearranging entirely the discussion of the results, for instance, selecting the most relevant ones and grouping them into subsections (i.e., the overall gap at the outbreak, the temporal evolution, the moderation, etc.).

Some minor points:

1) Descriptive figures 1, 2, and 3 were not visible in the version for the review and therefore it was not possible to comment.

2) Think about the possibility of including interactions in the main text potentially in a graphical way to simplify the understanding of this large set of results.

3) Avoid using causal language as “effect” when discussing the results.

4) Given the repeated cross-sectional design of the analyses, I would avoid stressing the “longitudinal component” of your contribution. The term diachronic seems to be more appropriate.

5) On the previous point, why did you not exploit the longitudinal nature of the panel? Other studies have done something similar in other contexts (i.e., Zamberlan et al. 2022, although not acknowledged in the literature review).

Zamberlan, A., Gioachin, F., & Gritti, D. (2022). Gender inequality in domestic chores over ten months of the UK COVID-19 pandemic. Demographic Research, 46, 565-580.

6) The section on limitations does not adequately address limitations, other than the fact that it is a study of only one country (which, in my opinion, is not a limitation if the contribution is properly contextualized). Please elaborate further.

Reviewer #2: The study is interesting and has a potential to make a contribution to better understand of the development of gender inequalities in paid work, childcare and wellbeing throughout the pandemic, especially in later stages. It identifies that there is a need for greater evidence on this and attempts to fill in the evidence gap. However, the manuscript requires major revisions to meet journal requirements.

1. Representativeness/selectiveness of the analytical sample

Main concern relates no effort made to compare representativeness of their “final analytical sample” of the underlying population of workers across the time points analysed. Authors state that a representative, probability-based panel was derived, but it is not clear how the response rates, changes in study design as well as subsequence exclusions affected the representatives of the sample and the subsequent conclusions.

Table 1 shows that at some waves less than a half of the sample made it to their analytical sample, but these proportions vary substantially over time. This is concerning for two main reasons. Firstly, previous studies show that women dropped out from employment at higher rates than men in the early stages of the pandemic. Given the sample is selected based on who remains in paid employment, such inequalities are neglected. Secondly, cross sectional comparison over time can only be meaningful if the same individuals are compared. If this is not possible efforts should be made to ensure that the final analytical sample has the same characteristics over time points compared and is representative of the population of interest. Table S2 compares the samples in terms of the proportion of female but I would be useful to know how the samples compare in terms of the other covariates as well.

2. Missing data strategy

The authors also state that respondents with missing values were excluded, which would further impact how representative the analytical sample is and therefore how meaningful a comparison over time can be made. Perhaps a more elaborate missing data strategy, such as multiple imputations, should be considered?

3. Contextualisation

The paper is lacking contextualising both in terms of situating the data in the timeline as well as situating the results in the existing literature. Authors present the timeline in S1, but it is not clear to what extent these restrictions could impact on the results. For example, were the restrictions on work different for those in essential and non-essential occupations? The impact of the restriction on work should feature in the discussion of the findings more explicitly. Similarly, the paper identifies several important international studies, but these do not feed through to the discussion section – are the results similar or different to what previous studies have found? If so, why/why not?

4. Comprehensiveness of the work location outcome variable

It is not clear whether workers were allowed to work from home “by choice”. The variable distinguishes between those who: work from home; work from workplace “by choice”; work from workplace “due to the nature of the work”. At the same time the work autonomy variable allows the choice in terms of work location. This appears inconsistent. Furthermore, previous studies on gender inequality suggest that choices men and women make may reflect internalised social norms. For example, men may be more concerned than women with “breadwinner status” than reputational damage related to neglecting childcare responsibilities, which would affect their behaviours. This should be recognised.

Furthermore, “working partially form home” is shown only in the Figure 1, but not modelled as separate category, which appears inconsistent.

5. Floor/ceiling effects

In the analyses of childcare responsibilities, authors define whether respondent are doing more/less compared to before the pandemic. At the same time authors state that division of childcare in the Netherlands was significantly gendered prior to the pandemic, with mothers taking on a much greater role in childcare than fathers. This implies that it may have not been possible for mothers to do even more. Were the ceiling and floor effects of such measurement considered? In the discussion it is also sometimes not clear whether the respondents are doing more/less than they used to or more/less than their partners.

There are also several minor issues:

6. The specific hypothesis and research questions should be made more explicit in the introduction.

7. There are also issues with reporting self-perceived measures retrospectively, as the subjective perception may change over time or people may simply forget/romanticise/misinterpret. These should be acknowledged.

8. More rationale/explanation for the definition of essential occupation is needed. Are healthcare workers or teachers not considered as essential?

9. If possible, it would be useful to control for the age of children in the household, as younger children might need more attention.

Once these issues have been addressed, I would recommend the article for publication.

6. PLOS authors have the option to publish the peer review history of their article (what does this mean?). If published, this will include your full peer review and any attached files.

Reviewer #1: No

Reviewer #2: No

---

## [Author Response · Author response to Decision Letter 0]

30 Jan 2024

Dear Florian Fischer, 

We are pleased to read that you and the two anonymous reviewers feel our manuscript has merit. We further appreciate the opportunity to improve the manuscript based on the comments provided. In this rebuttal letter, we respond to each point raised by yourself and the reviewers. Together with our rebuttal letter, we have included a marked-up copy of our manuscript that highlights changes made to the original version (Revised Manuscript with Track Changes) and an unmarked version of our revised paper without tracked changes (Manuscript).

We feel the changes have substantially improved the manuscript and we look forward to hearing from you and the reviewers in due course. 

Kind regards, also on behalf of my co-authors,

Mara Yerkes

Academic editor comments:

Response: We have carefully edited the manuscript to ensure it follows all style requirements. This includes a check of all headings and removing the short title from the title page. We also now first mention the asterisk followed by the pilcrow symbol. We have also uploaded all Figures to PACE to ensure compliance with PLOS ONE formatting.

Response: There are legal restrictions to sharing our data publicly. The data used for our analyses are owned by the LISS panel and stored in the LISS data archive. Any person interested in replicating our analyses can request access to the LISS data archive and download the data. Subsequently, all steps necessary to reproduce the analyses (i.e., the syntax files) with these data files are outlined in .txt files, all of which will be available on the project website www.cogisnl.eu/dataandsyntax following acceptance of the article for publication. 

3. We notice that your supplementary figures are uploaded with the file type 'Figure'. Please amend the file type to 'Supporting Information'. Please ensure that each Supporting Information file has a legend listed in the manuscript after the references list.

Response: We have amended the title of all supplementary figures and tables to ‘Supporting Information’. We have also ensured that each file has a legend listed in the manuscript after the references list.

Reviewer #1: This study addresses the diachronic evolution of gender differences in work arrangements, childcare division, and work-life balance.

The descriptive findings extend the existing knowledge by covering approximately two years since the outbreak of the pandemic emergency. The topic is highly relevant for the literature addressing gender inequality and COVID-19-related inequalities, and the paper proposes to make a significant contribution to these readerships. However, major revisions are currently required to meet the publication criteria.

Response: We thank the reviewer for recognizing the significant contribution our article can make and its relevance for the literature.

My major points are:

1) INTRODUCTION AND RESEARCH QUESTION: I personally appreciate the review of existing studies on the topic produced over the last few years, how the existing evidence is critically discussed, and how the question about the evolution of existing disparity over the 2020-2022 period emerges. However, the hypotheses appear poorly motivated and are built upon previous works, primarily from the UK. Although the article does not seem to engage in a detailed theoretical discussion, I believe that the authors should discuss the theoretical mechanisms behind the discussed expectations and the theoretical roles of the mentioned moderators. Furthermore, these motivated expectations require adequate contextualization given the country under scrutiny.

Response: In re-reading the article, we agree with the reviewer (and Reviewer 2, see point 6 below) that our research question and hypotheses required clarification. We indeed do not provide a detailed theoretical discussion here, for reasons of brevity but also given the broad empirical scope and aim of the article. We now refer to other publications in which potential theoretical mechanisms are discussed in more detail, while opting for a short (but in our eyes sufficient) discussion of potential theoretical mechanisms behind our expectations for the current analyses as well as the moderators. These theoretical discussions are now provided together with greater contextualization for the Dutch case in a new section (The Potential for Sensemaking and the Dutch Context, p. 7). The additional information on the Dutch context is also provided to provide greater contextualization in relation to our data collection (see point 3, Reviewer 2). Based on these theoretical discussions, we feel it is prudent to note that we are exploring these relationships throughout the pandemic, rather than testing a specific hypothesis. We now state this in the text (see p. 9).

2) NOT CLEAR SAMPLE STRATEGY: I found it difficult to understand who composed the sample. On page 7, the authors specify that the survey was handed to LISS panel members with at least one working member and a co-resident minor child. However, in the analyses, having or not a child became a control variable (or parenthood as a moderator). Descriptive statistics for the variables of interest were missing. Moreover, if the sample can or cannot have children, the analysis of the childcare division necessarily has a sample different from that of the other analyses. This should be discussed and a description of who composes the different samples should be provided.

Response: We thank the reviewer for this comment. We have now clarified the distinction between the original sample (wave 1) and the extension of this sample in wave 2 to include households without minor co-resident children (see pp. 12-13). Furthermore, we have provided descriptive statistics for all variables of interest (see S2-S5 tables). 

3) SELECTION IN SAMPLE: The sampling criteria and two outcome variables require the respondents to be in paid employment. As many of the COVID-19 implications, particularly on gender inequalities, have also occurred by affecting (un-)employment chances, you specifically make a selection of individuals that are still employed in each wave. This selection is not incorrect, but it should also be discussed in relation to the composition of this sample between men and women. Finally, is this selection too strong for the analysis of childcare divisions?

Response: We thank the reviewer for raising this point. Indeed, we recognize that for two outcome variables (work location and work-life balance), respondents indeed need to be in paid employment. We also agree that in many country contexts, gender inequalities have been shaped by (un-)employment chances. In our case country of the Netherlands, however, unemployment (particularly for women) was quite low during the pandemic, likely due to the presence of extensive government measures to protect workers. Additionally, respondents in our sample were asked to indicate whether they were out of work due to the pandemic; only a very small percentage (1-3% from waves 1 through 4) of respondents indicated yes. We reference relevant literature and highlight these points in our Methods section (p. 13) to demonstrate that pandemic-based unemployment does not significantly influence our sampling strategy. 

Extending on from this, as the occurrence of COVID-19 related unemployment is so low in our sample, and because parents can be partnered with someone who is unemployed or otherwise not working, we do not feel that our selection is too strong for the analysis of childcare divisions. The table with information on the selection for our dependent variables can be found in the Supplemental Information S3 Table.

4) OUTCOME VARIABLES: Although I like the pre-post operationalization of the outcome variables (i.e. what is the difference with the pre-pandemic period), it would be anyway interesting to look at the absolute values of each wave. This is especially true for childcare, as children may grow over two or more years, and changes in the degree of childcare involvement may be driven by the aging of the child. Moreover, one may also consider the construction of a “within-couple disparity” measure for childcare, where the outcome becomes an imbalance (or changes in the balance) between the two partners.

Response: We thank the reviewer for these suggestions. We agree that particularly for childcare, changes in the degree of childcare involvement could be driven by the aging of the child. In other publications (anonymized for the purposes of review), we have indeed looked at the absolute values of each wave as an outcome variable. However, given the complexity of the current manuscript, and the request (point 6 below) to simplify the manuscript rather than add more analyses, we have decided to refer to our work looking at each wave rather than include it here. Similarly, while a “within-couple disparity” measure would be a useful addition if we were looking solely at the division of childcare, as we are focused on a broader analysis of gender differences throughout the pandemic, we do not include it here. 

5) CONTROL VARIABLES: In the analyses, other theoretically driven covariates should be considered. The age of the (youngest) child, for instance, is indicative of the required childcare, the number of children also corrects for differential needs between families, and the biological sex of the child also matters for parent-child interaction. Moreover, the partners’ characteristics and health conditions should be considered. Finally, childcare involvement (and particularly paternal childcare involvement) should also be regarded as a control variable in the study of work-life balance (if all the sampled individuals have at least one child, but indeed this is not clear). For men, it may be easier to combine work and life duties if they do not carry-on childcare activities at all, so men with similar childcare burdens across time should be compared.

Response: We thank the reviewer for providing additional suggestions for control variables. We have now included additional partner characteristics (in addition to whether or not the partner has an essential occupation, we now also include partner’s work location). We include the age of the youngest child for our analyses of the division of childcare and the robustness checks on the sub-sample of parents for the work-life balance outcome (see below; S32-S37 Tables). 

Although we agree that the biological sex of the child could be important to include as a control for parent-child interaction, given our focus on gender equality more broadly, we do not include this information here. Additionally, we unfortunately do not have information on health conditions across all data points (such information is only available as part of the core studies of LISS, which are collected on an annual basis). We have therefore chosen to not include them here as a control. We also discuss the exclusion of potential con founders in the Limitations and Future Research section of the Discussion (see p. 31-33). Lastly, in our analyses of work-life balance, the sample includes respondents both with and without children. We have now run robustness checks on the sub-sample of parents in order to allow us to include childcare involvement as a covariate when analyzing the outcome of work-life balance. We have chosen to not include an additional interaction between gender and the level of childcare involvement as that is not the focus of our empirical study. Rather, we maintain our focus on the three moderators of parenthood, essential occupation and education. We report our robustness checks at the end of the Results section (see p. 27). 

6) RESULTS: Here is the major point of the review. While going through the paper, I felt that there were simply too many aspects under consideration that hampered a smooth reading and understanding of the contribution. First, the three outcomes are only marginally related to each other, sometimes referring to different reference populations, and in the end, they appear to be chosen more for convenience than for specific theoretical reasons. Second, these outcomes are with multiple categories. Even though the multinomial approach is the correct one, the amount of information to be commented on for each analysis (also considering the two/three moderators) is overwhelming, and I took several readings to grasp them. Alternative operationalizations may help simplify the analyses and communication of the results. Third, the discussion of the results is currently not helpful in simplifying the provided information. In the current version, each result is discussed didactically, without any underlying narrative. Besides reconsidering the overly large focus of the contribution, I would suggest rearranging entirely the discussion of the results, for instance, selecting the most relevant ones and grouping them into subsections (i.e., the overall gap at the outbreak, the temporal evolution, the moderation, etc.).

Response: We thank the reviewer for the helpful suggestion on how to re-organize our results section. To address all three of the points raised (6a: relationship between the three outcomes, 6b: the amount of information being communicated, and 6c: the discussion of the results), we have thoroughly revised the Results section in line with the suggestions of the reviewer. We start by presenting pre-pandemic differences between men and women, followed by the temporal evolution of differences (phase-specific), focused on what changed and when. Lastly, we present our moderated analyses focusing on potential drivers of these gender differences. As the re-organization of our Results section was so extensive, we have done this without track changes (see pp. 21-27). The same is true for the Discussion section (see pp. 27-33). We have also revised the front end of the paper to clarify how these three outcomes relate to each other more broadly, in relation to gender inequalities in work, care and wellbeing (see pp. 4-11; point 6a). 

Some minor points:

1) Descriptive figures 1, 2, and 3 were not visible in the version for the review and therefore it was not possible to comment.

Response: We apologize for this oversight. The 

---

## [Decision Letter · Decision Letter 1]

10 Apr 2024

Gender inequality in work location, childcare and work-life balance: phase-specific differences throughout the COVID-19 pandemic

PONE-D-23-16207R1

Dear Dr. Yerkes,

We’re pleased to inform you that your manuscript has been judged scientifically suitable for publication and will be formally accepted for publication once it meets all outstanding technical requirements.

Kind regards,

Kyoung-Sae Na, M.D., Ph.D.

Academic Editor

PLOS ONE

Additional Editor Comments (optional):

Reviewers' comments:

Reviewer's Responses to Questions

**Comments to the Author**

1. If the authors have adequately addressed your comments raised in a previous round of review and you feel that this manuscript is now acceptable for publication, you may indicate that here to bypass the “Comments to the Author” section, enter your conflict of interest statement in the “Confidential to Editor” section, and submit your "Accept" recommendation.

Reviewer #1: All comments have been addressed

2. Is the manuscript technically sound, and do the data support the conclusions?

Reviewer #1: Yes

3. Has the statistical analysis been performed appropriately and rigorously? 

Reviewer #1: Yes

4. Have the authors made all data underlying the findings in their manuscript fully available?

Reviewer #1: Yes

5. Is the manuscript presented in an intelligible fashion and written in standard English?

Reviewer #1: Yes

6. Review Comments to the Author

Reviewer #1: In preparing the current version of the manuscript, the Author(s) precisely followed the suggestions made by the reviewers, changing (even extensively) the structure and content of the text. Where the reviewers' suggestions were not entirely followed, the authors provided adequate justification in their answers, but above all in the text, highlighting the existing limitations of the study. Well done!

In light of this new version, I would recommend the article for publication.

7. PLOS authors have the option to publish the peer review history of their article (what does this mean?). If published, this will include your full peer review and any attached files.

Reviewer #1: No

---

## [Editor Report · Acceptance letter]

30 Apr 2024

PONE-D-23-16207R1 

PLOS ONE

Dear Dr. Yerkes, 

I'm pleased to inform you that your manuscript has been deemed suitable for publication in PLOS ONE. Congratulations! Your manuscript is now being handed over to our production team.

Kind regards, 

on behalf of

Dr. Kyoung-Sae Na 

Academic Editor

PLOS ONE